# Moving Target Indication for Dual-Channel Circular SAR/GMTI Systems

**DOI:** 10.3390/s20010158

**Published:** 2019-12-25

**Authors:** Laihe Wang, Yueli Li, Wu Wang, Daoxiang An

**Affiliations:** 1College of Electronic Science and Technology, National University of Defense Technology, Changsha 410073, China; wanglaihe94@163.com (L.W.); wangwu163@163.com (W.W.); daoxiangan@nudt.edu.cn (D.A.); 2Unit 13, the No. 92493 Troop of PLA, Huludao 125001, China

**Keywords:** dual-channel circular synthetic aperture radar, moving target detection, channel registration, channel equalization, clutter suppression

## Abstract

In a dual-channel circular synthetic aperture radar (CSAR) and ground moving target indication (GMTI) system, the antenna baseline is not parallel with the flight path due to a yaw angle. The angle causes a varying group-phase shift between the dual-channel signals and therefore degrades the correlation between the image pair. Therefore, the group-phase shift needs to be removed before channel equalization. To resolve the problem, the interferometric phase term was deduced and analyzed based on the geometry of a dual-channel CSAR system. Then, the varying phase term with respect to the Doppler frequency and the varying group-phase shift over the range were compensated for in the channel registration. Furthermore, blind channel equalization, including two-dimensional calibration and amplitude equalization, was applied to eliminate the amplitude and residual phase differences between the channels. Finally, the amplitude image obtained using a displaced phase center antenna (DPCA) was multiplied by the phase image obtained with along-track interferometry (ATI) to detect moving targets. The experimental results verified the effectiveness of the method for both uniform and non-uniform clutter suppression.

## 1. Introduction

Synthetic aperture radar (SAR) combined with ground moving target indication (GMTI) [1] technology has been extensively studied in remote sensing. The circular SAR (CSAR), which employs the long-time and multi-angle observation of a scene, has the potential for moving target detection (MTD) [2]. Since a dual-channel SAR system has a larger spatial freedom, the clutter of the scene might be suppressed more effectively in it than in a single channel one [3,4]. However, the carrier yaw in the dual-channel CSAR geometry causes group-phase shift [5] between the CSAR image pair. Both the amplitude and phase differences affect the image correlation and might give rise to residual components after clutter suppression. The residual clutter has a negative impact on the performance of MTI methods including along-track interferometry (ATI) and displaced phase center antenna (DPCA) techniques [6,7]. For example, DPCA is sensitive to the channel imbalance [8,9,10].

Recently, GMTI based on a dual-channel SAR data has raised significant interest among researchers because of its advantages in clutter suppression. In Gao and Shi [11], a notch filter was capable of detecting ship targets from the image pairs obtained using a dual-channel ATI-SAR. However, the filter is only suited for ocean applications. In Shen et al. [3] and Uysal et al. [12], pixels in a series of images need to be sequenced to select the medium value, and then all the medium value pixels constitute the stationary clutter image. However, the algorithm increases the computational complexity significantly. Furthermore, the influence of the CSAR geometry is not extensively studied. In Wang et al. [13], we proposed a DPCA function to suppress the stationary clutter between a dual-channel CSAR image pair. However, the varying phase differences in the interferometric diagram were not well addressed. In this study, we deduced the phase differences of the dual-channel signals in the CSAR imaging geometry. We also proposed a phase factor to efficiently compensate for the varying phase differences between channels. Then, an entire MTI procedure, which included channel registration, channel equalization, and a DPCA plus ATI method, was utilized to suppress the clutter, as well as to detect moving targets.

The paper is organized according to the scheme of signal processing. Section 2 deduces the signal model based on the geometry of a dual-channel CSAR system and analyzes the phase variation related to the Doppler frequency and range. Section 3 introduces a modified MTI method and Section 4 demonstrates its validity with an X-band Gotcha challenge dataset. Finally, the conclusion is provided in Section 5.

## 2. Dual-Channel CSAR/GMTI Imaging Geometry

Figure 1 illustrates the imaging geometry for an airborne dual-channel CSAR/GMTI system. The symbols $A_{1}$ and $A_{2}$ represent the antenna phase centers of the radar system, and $A_{0}$ is the projection of $A_{1}$ onto the flight path. *R_B_*_1_ and $R_{B2}$ are the closest ranges between $A_{1}$ and $A_{2}$ and a point target P on the ground plane, respectively. Meanwhile, $R_{1}\left(t_{m};R_{B1}\right)$ and $R_{2}\left(t_{m};R_{B2}\right)$ are the instantaneous slant ranges at the slow time $t_{m}$.

Without loss of generality, we consider that the carrier is flying along a straight line with a velocity v during the imaging event. Hence, the instantaneous slant ranges can be approximately expressed based on the Taylor series expansion:(1)$$\begin{array}{ll} R_{1}\left(t_{m};R_{B1}\right) & =\sqrt{{\left(vt_{m}+d\cos\theta \right)}^{2}+{\left(R_{B2}\cos\phi +d\sin\theta \right)}^{2}+h^{2}}\\ & \approx R_{B1}+\frac{vd\cos\theta }{R_{B1}}t_{m}+\left[1-\frac{{\left(d\cos\theta \right)}^{2}}{R_{B1}{}^{2}}\right]\frac{v^{2}}{2R_{B1}}t_{m}{}^{2} \end{array},$$
(2)$$R_{2}\left(t_{m};R_{B2}\right)=\sqrt{{\left(vt_{m}\right)}^{2}+R_{B2}{}^{2}}\approx R_{B2}+\frac{v^{2}t_{m}{}^{2}}{2R_{B2}},$$
where the symbol *d* denotes the equivalent distance between two phase centers, θ denotes the yaw angle between the antenna baseline and the flight path, ϕ denotes the grazing angle between the slant-range $R_{B2}$ and the ground plane, and *h* denotes the flight height of the carrier. According to the imaging geometry, $R_{B1}$ can be defined as:(3)$$\begin{array}{ll} R_{B1} & =\sqrt{{\left(R_{B2}\sin\phi \right)}^{2}+{\left(R_{B2}\cos\phi +d\sin\theta \right)}^{2}+{\left(d\cos\theta \right)}^{2}}\\ & =\sqrt{R_{B2}^{2}+d^{2}+2R_{B2}d\cos\phi \sin\theta } \end{array}.$$

After the range compression, the dual-channel signals are given by:(4)$$s_{1}\left(t,t_{m}\right)=\sigma_{1}p_{r}\left[t-\frac{2R_{1}\left(t_{m};R_{B1}\right)}{c}\right]\exp\left[-jk_{c}2R_{1}\left(t_{m};R_{B1}\right)\right],$$
(5)$$s_{2}\left(t,t_{m}\right)=\sigma_{2}p_{r}\left[t-\frac{2R_{2}\left(t_{m};R_{B2}\right)}{c}\right]\exp\left[-jk_{c}2R_{2}\left(t_{m};R_{B2}\right)\right],$$
where *t* represents the fast time, $\sigma_{1}$ and $\sigma_{2}$ are the complex constants that represent the signal amplitudes, $p_{r}$ denotes the range impulse response function, and $k_{c}=2\pi f_{0}/c$ is the space wave number corresponding to the center frequency $f_{0}$ of the transmitted signals.

The range Doppler spectrum is obtained by applying the principle of stationary phase (POSP) to Equation (4):(6)$$S_{1}(t,f_{a})=\kappa_{1}p_{r}\left[t-\frac{2R_{1}\left(f_{a};R_{B1}\right)}{c}\right]\exp\left\{-j\frac{4\pi R_{B1}}{\lambda }\left[1-\frac{{\left(\frac{\lambda f_{a}}{2}+v\sin\gamma \right)}^{2}}{2v^{2}\cos^{2}\gamma }\right]\right\},$$
where $\kappa_{i}$ (*i* = 1, 2) represent the amplitude of the signals, $f_{a}$ denotes the Doppler frequency, and λ represents the wavelength of $f_{0}$. γ is the angle between $PA_{0}$ and $PA_{2}$, which is given by:(7)$$\sin\gamma =\frac{d\cos\theta }{R_{B1}}\approx \frac{d\cos\theta }{R_{B2}}.$$

Similarly, the range-Doppler spectrum of Equation (5) is expressed by:(8)$$S_{2}(t,f_{a})=\kappa_{2}p_{r}\left[t-\frac{2R_{2}\left(f_{a};R_{B2}\right)}{c}\right]\exp\left[-j\frac{4\pi R_{B2}}{\lambda_{0}}\left(1-\frac{\lambda^{2}f_{a}^{2}}{8v^{2}}\right)\right].$$

Multiplying the normalized spectrum ${\bar{S}}_{2}\left(\tau ,f_{a}\right)$ by the normalized conjugated spectrum ${\bar{S}}_{1}{}^{*}\left(\tau ,f_{a}\right)$, the phase difference between the dual-channel signals can be calculated using the small-angle approximations, as follows:(9)$$\begin{array}{ll} ∆\varphi & =\frac{4\pi R_{B2}}{\lambda }\left(\frac{\lambda^{2}f_{a}^{2}}{8v^{2}}-1\right)+\frac{4\pi R_{B1}}{\lambda }\left(1-\frac{{\left(\frac{\lambda f_{a}}{2}+v\sin\gamma \right)}^{2}}{2v^{2}\cos^{2}\gamma }\right)\\ & \approx \frac{4\pi d\sin\theta \cos\phi }{\lambda }-\frac{2\pi d\cos\theta }{v}f_{a}-\frac{2\pi d\sin\theta \cos\phi }{\lambda }{\left(\frac{f_{a}}{f_{aM}}\right)}^{2} \end{array}$$
where $f_{aM}=2v/\lambda$ denotes the maximal Doppler shift. The first term in Equation (9) indicates that a phase difference varies linearly with cosϕ. The second term is linearly proportional to the Doppler frequency. The third term is a quadratic term relevant to the Doppler frequency.

The amplitude and phase differences between the dual-channel signals cause an image mismatch that hinders the detection of moving targets. To tackle the problem, we need to manage these differences to suppress the clutter.

## 3. CSAR GMTI Method

Figure 2 presents the flowchart of a dual-channel CSAR GMTI method. Usually, a channel registration is utilized to align the dual-channel signals after range compression. First, the varying phase differences and group-phase shift between the dual-channel signals are compensated for by using a phase factor in the channel registration. Second, the amplitude and residual phase differences are eliminated by using channel equalization, which includes two-dimensional calibration and amplitude equalization. Third, a back-projection (BP) algorithm is utilized for image formation. Finally, moving targets in the image pair are detected by using a modified weighted DPCA method.

### 3.1. Channel Registration

After range compression and conjugate multiplication in the Doppler domain, the interferometric phase diagram is obtained. According to Equation (9), the first phase term varies with cosϕ, indicating a group-phase shift over the range direction. The second phase term varies linearly with the Doppler frequency. Therefore, the low order phase difference in a range bin can be recast as:(10)$$∆\varphi_{linear}=\frac{4\pi d\sin\theta \cos\phi }{\lambda }-\frac{2\pi d\cos\theta }{v}f_{a}=D+kf_{a},$$
where D denotes the group-phase shift of a range bin and k denotes the slope of the interferometric phase on the Doppler axis in the range bin. Thus, the linear phase compensation factor is constructed using:(11)$$H\left(f_{a}\right)=\exp\left(-j(D+kf_{a})\right).$$

As precise angle measurements are not available due to the turbulence during the flight, it is not feasible to calculate the low-order phase difference in each range bin. To resolve the problem, we estimate the phase difference of range bins from the interferometric diagram. The steps for phase difference estimation are as follows:(1)Estimating the slope of the interferometric phase in the diagram: The varying phase curve with respect to the Doppler frequency in each range bin is extracted. Then, the slope of each curve is estimated by using curve fitting. According to Equation (9), the slope in different bins is identical. Thus, the mean value of the estimates is considered to be the real slope in the entire diagram.(2)Measuring the phase shift in each range bin: The zero Doppler point on the abscissa of the interferometric phase diagram taken to be a reference point. Then, the distance from the reference point to the intersection point where the phase curve cuts the Doppler axis is measured. The measured value is the phase shift of the range bin.

After multiplying Equation (11) by Equation (6), the group-phase shift and linear varying phase term between the dual-channel signals are removed. However, according to Equation (9), there is a rudimental quadratic phase term in the interferometric diagram. Furthermore, amplitude difference between the dual-channel signals also needs to be addressed.

### 3.2. Channel Equalization

In the two-dimensional frequency domain, the dual-channel signals after the channel registration can be simplified to:(12)$$Z_{1}\left(f_{a},f_{r}\right)≅a\left(f_{a}\right)h_{1}\left(f_{r}\right)D_{1}\left(f_{a}\right),$$
(13)$$Z_{2}\left(f_{a},f_{r}\right)≅a\left(f_{a}\right)h_{2}\left(f_{r}\right)D_{2}\left(f_{a}\right),$$
where $f_{r}$ represents the range frequency, $h_{i}\left(f_{r}\right)$ (*i* = 1, 2) are the channel transfer functions, $a\left(f_{a}\right)$ denotes the amplitude of the signal, and $D_{i}\left(f_{a}\right)$ (*i* = 1, 2) denote the phase items associated with the slant range.

Blind channel equalization is utilized to address the channel imbalance, including the residual phase differences and amplitude differences. The ratio between the dual-channel signals can be defined as:(14)$$\frac{Z_{1}\left(f_{a},f_{r}\right)}{Z_{2}\left(f_{a},f_{r}\right)}≅\frac{h_{1}\left(f_{r}\right)}{h_{2}\left(f_{r}\right)}\frac{D_{1}\left(f_{a}\right)}{D_{2}\left(f_{a}\right)}.$$

Obtaining the powers of the channel difference and calculating the minimum integration over both the range frequency and Doppler frequency axes gives:(15)$$\underset{h_{1,2}\left(f_{r}\right),D_{1,2}\left(f_{a}\right)}{\min}{\int \left|Z_{1}\left(f_{a},f_{r}\right)-Z_{2}\left(f_{a},f_{r}\right)h_{1,2}\left(f_{r}\right)D_{1,2}\left(f_{a}\right)\right|}^{2}df_{a}df_{r},$$
where $h_{1,2}=\frac{h_{1}}{h_{2}}$ denotes the imbalance ratio caused by the channel transfer functions and $D_{1,2}=\frac{D_{1}}{D_{2}}$ represents the residual phase difference. The values are given by:(16)$$h_{1,2}\left(f_{r}\right)=\frac{\int D_{1,2}{\left(f_{a}\right)}^{*}Z_{2}{\left(f_{a},f_{r}\right)}^{*}Z_{1}\left(f_{a},f_{r}\right)df_{a}}{\int {\left|Z_{2}\left(f_{a},f_{r}\right)\int D_{1,2}\left(f_{a}\right)\right|}^{2}df_{a}},$$
(17)$$D_{1,2}\left(f_{a}\right)=\frac{\int h_{1,2}{\left(f_{r}\right)}^{*}Z_{2}{\left(f_{a},f_{r}\right)}^{*}Z_{1}\left(f_{a},f_{r}\right)df_{r}}{\int {\left|Z_{2}\left(f_{a},f_{r}\right)\int h_{1,2}\left(f_{r}\right)\right|}^{2}df_{r}},$$
where $Z_{2}^{\left(n,i\right)}\left(f_{a},f_{r}\right)$ can be iteratively solved using:(18)$$\begin{array}{l} Z_{2}^{\left(n,1\right)}\left(f_{a},f_{r}\right)=Z_{2}^{\left(n-1,2\right)}\left(f_{a},f_{r}\right)\frac{{\int Z_{2}^{\left(n-1,2\right)}\left(f_{a},f_{r}\right)}^{*}Z_{1}\left(f_{a},f_{r}\right)df_{a}}{\int {\left|Z_{2}^{\left(n-1,2\right)}\left(f_{a},f_{r}\right)\right|}^{2}df_{a}}\\ Z_{2}^{\left(n,2\right)}\left(f_{a},f_{r}\right)=Z_{2}^{\left(n,1\right)}\left(f_{a},f_{r}\right)\frac{{\int Z_{2}^{\left(n,1\right)}\left(f_{a},f_{r}\right)}^{*}Z_{1}\left(f_{a},f_{r}\right)df_{r}}{\int {\left|Z_{2}^{\left(n,1\right)}\left(f_{a},f_{r}\right)\right|}^{2}df_{r}} \end{array},$$
where *n* is the number of iterations, $Z_{2}\left(f_{a},f_{r}\right)$ is the initial value $Z_{2}{}^{\left(0,1\right)}(f_{a},f_{r})$. $Z_{2}^{\left(n,i\right)}\left(f_{a},f_{r}\right)$ (*n* = 1, 2, …; *i* = 1, 2) is the output of the *n*th iteration for the second channel. Experiment results show that the correlation coefficient remains constant while n≥3. Since channel equalization is implemented in the two-dimensional frequency domain, it is called 2D calibration.

2D calibration is very successful at eliminating the amplitude and residual phase differences of the main lobe between the dual-channel signals, while the amplitude differences of the side lobes remains large. Therefore, amplitude equalization is utilized to remove the amplitude differences in the side-lobes. Since the antenna beam covers the entire range unit at the same time, the amplitude difference between the two channels is assumed to be constant for different range cells. The amplitude error factor *A* is the square root of the ratio of the power spectrum in the Doppler domain, as follows [14]:(19)$$A^{2}=\frac{PSW\left[S_{2}\left(f_{a}\right)\right]}{PSW\left[S_{1}\left(f_{a}\right)\right]}.$$

According to Equation (19), *A* is utilized to compensate for the amplitude difference in the side-lobes.

As channel registration and channel equalization have little influence on the imaging process, we will not discuss the BP algorithm in this paper due to a length limitation.

### 3.3. Weighted DPCA Clutter Suppression

In Zhao and Li [6], a weighted ATI method is proposed to manage the uniform clutter using a detection function $\xi_{1}={\left|x_{1}-x_{2}\right|}^{2}\left(x_{1}x_{2}^{*}\right)$. In Shi [7], a detection method that combines DPCA and ATI techniques is proposed to suppress the clutter by using a detection function $\xi_{2}=\left|x_{1}x_{2}^{*}\right|\left(1-\cos(∆\varphi )\right)$, where ∆ϕ is the phase angle of the ATI. In both detection functions, $x_{1}$ and $x_{2}$ represent the dual-channel complex images. However, both methods fail to suppress the non-uniform clutter.

Note that the non-uniform clutter has a wider bandwidth than the uniform clutter. To suppress the non-uniform clutter, we propose a weighted DPCA method by constructing a detection function, as follows:(20)$$W_{DPCA}=\left|x_{1}-x_{2}\right|\left[1-\cos\left(∆\varphi \right)+\left|\sin\left(∆\varphi \right)\right|\right].$$

The proposed detection function has a wider bandwidth compared to the former methods.

## 4. Experimental Results and Discussions

The GMTI method was validated by using a dataset collected by a three-channel, X-band CSAR system. The X-band Gotcha challenge dataset is published by the US Air Force Laboratory (AFRL). The 11th-second data are utilized to validate the effectiveness of the proposed method. The processing results are presented and discussed step by step in the following sub-sections.

### 4.1. Channel Registration Results Using the Gotcha Challenge Dataset

The interferometric phase diagram of the dual-channel signals is illustrated in Figure 3a, where interferometric phase varied with the Doppler frequency $f_{a}$, which confirmed the second phase term of Equation (9). Moreover, the phase varied with the slant range, i.e., the group-phase shift. The procedures for the channel registration were manipulated as follows.
(1)Estimating the slope of the interferometric phase: As shown in Figure 3a, in the experiment, the curve fitting results of all 384 range cells were almost identical. The phase curves of the 130th, 260th, and 382nd range bins in Figure 3a are illustrated in Figure 4. As can be seen in Figure 4a, the slopes of the phase curves were identical, while their phase shifts from the zero Doppler point were different. In the interferometric phase diagram, the mean value of the slope was k=−0.01205.(2)Extracting the phase shift of each range bin: Figure 4b presents the partially enlarged details of the phase curves. The zero Doppler point is marked as the reference point RP. D1, D2, and D3 were the phase shifts in the 130th, 260th, and 382nd range bins, respectively.(3)Phase error compensation: Figure 3b shows the interferometric phase diagram, where only the linear phase term varying over the Doppler frequency was compensated for. The residual phase difference varying over the range is clearly visible in Figure 3b. Figure 3c presents the interferometric phase diagram after multiplying by the linear phase compensation factor in Equation (11). In contrast to Figure 3b, the phase variation over the range disappeared, indicating the group-phase shift was eliminated. In addition, the correlation coefficient of the dual-channel signals increased from 0.3677 to 0.9523. The correlation coefficient is defined as $R\left(x_{1},x_{2}\right)=C\left(x_{1},x_{2}\right)/\sqrt{C\left(x_{1},x_{1}\right)C\left(x_{2},x_{2}\right)}$, where $C\left(x_{i},x_{j}\right)$ is the covariance matrix between $x_{i}$ and $x_{j}$. Although most phase differences between channels were compensated for, a residual phase variation over the Doppler frequency was also observed, which implies a residual phase difference.

### 4.2. Channel Equalization Results Using the Gotcha Challenge Dataset

After the channel registration, channel equalization was implemented to address the channel imbalance. By averaging the dual-channel signals over the range in the Doppler domain, the distributions of the signal amplitude before and after the 2D calibration were compared, as shown in Figure 5a,b. In contrast to Figure 5a, Figure 5b shows the amplitude difference in the main lobe was corrected after the 2D calibration. However, the amplitude difference in the side-lobes was obvious. Then, according to Equation (19), the amplitude error factor *A* was used to compensate for the amplitude difference in the side-lobes. Figure 5c demonstrates that the amplitude equalization successfully removed the difference in the side-lobes.

The statistical histogram of the interferometric phase after channel registration is illustrated in Figure 6a. The peak value of the angle was situated near 345°, which indicates an angle error of about −15°. Since the two signals have the same round-trip range after channel registration, the peak value should be located at 0° in an ideal case. Different channel transfer characteristics cause the angle error, and it affects the clutter cancellation. Therefore, channel equalization is critical for removing the phase error. After performing the channel equalization mentioned in Section 2, the new statistical histogram of the interference phase is illustrated in Figure 6b, where the statistical peak of the ground clutter at 0° indicates that the angle error was removed. Plus, the coherent coefficient between the two signals reached 0.9872.

### 4.3. Weighted MTI Clutter Suppression Results Using the Gotcha Challenge Dataset

Figure 7a presents the single-channel SAR image processed using the BP algorithm. In the image, a vehicle target is marked in a red box. The scenario contains both uniform and non-uniform clutter distribution regions. The clutter suppression results using $\xi_{1}$ from Zhao and Li [6] and $\xi_{2}$ from Shi [7] are shown in Figure 7b and Figure 7c, respectively. As it can be seen, the non-uniform clutter generated by buildings remained strong after the clutter suppression. It implies that both methods could suppress the uniform clutter effectively but failed to suppress the non-uniform clutter.

The MTI result of our proposed weighted DPCA method is presented in Figure 7d. As shown in the image, both the uniform and non-uniform clutters were suppressed. The moving target was also well-preserved.

The signal-clutter ratio (SCR) of the region of interest (ROI) in an image can be defined as [3]:(21)$$SCR=10\log\frac{\mu_{s}}{\mu_{c}},$$
where $\mu_{s}$ is the maximum intensity of the target signature in the ROI and $\mu_{c}$ is the maximum intensity of the surrounding clutter area. The ROI is centered around the target and the surrounding clutter area is three times as long and wide as the target size.

The SCR of the clutter suppression results manipulated with $\xi_{1}$, $\xi_{2}$, and $W_{DPCA}$ in Figure 7b, Figure 7c, and Figure 7d were 0.34 dB, 2.56 dB, and 5.20 dB, respectively. It is clear that the weighted DPCA method improved the performance for clutter suppression.

Figure 8 further compares the clutter suppression results of the three methods. The curves are the average amplitude curves of all the range cells. The blue and green curves in Figure 8 represent the amplitude curves processed using $\xi_{1}$ and $\xi_{2}$, respectively. The level of the moving target in both curves was lower compared to the clutter level of buildings in the center. This may cause false targets in the target indication. The red curve of the clutter processed with $W_{DPCA}$ demonstrates that the energy of the stationary clutter was effectively suppressed.

## 5. Conclusions

For the dual-channel CSAR/GMTI system, the image geometry had a significant impact on the moving target indication. To solve the problem, we proposed a modified GMTI method that included channel registration, channel equalization, and clutter suppression. In particular, the group-phase shift caused by the CSAR geometry were deduced and compensated for in the channel registration. Channel equalization was utilized to compensate for the channel imbalance, including the amplitude difference and residual phase difference. Furthermore, a weighted DPCA combined with ATI was proposed to suppress the uniform clutter, as well as the non-uniform clutter. The real data experiment validated the effectiveness of our proposed method.

Note that the proposed method utilized the dual-channel signals in the CSAR system to detect moving targets, which means that the correlation between the dual-channel signals was critical for performance. The correction of the amplitude and phase differences was manipulated in the signal region. Therefore, only an image pair needed to be utilized in the GMTI method, which improved the efficiency. Furthermore, experimental results show that the method could simultaneously remove the uniform and non-uniform clutters effectively. Although the method demonstrates a better performance using the Gotcha challenge dataset, the feasibility for a low-frequency CSAR system with a large image aperture needs to be verified in the future.

## Figures and Tables

**Figure 1 sensors-20-00158-f001:**
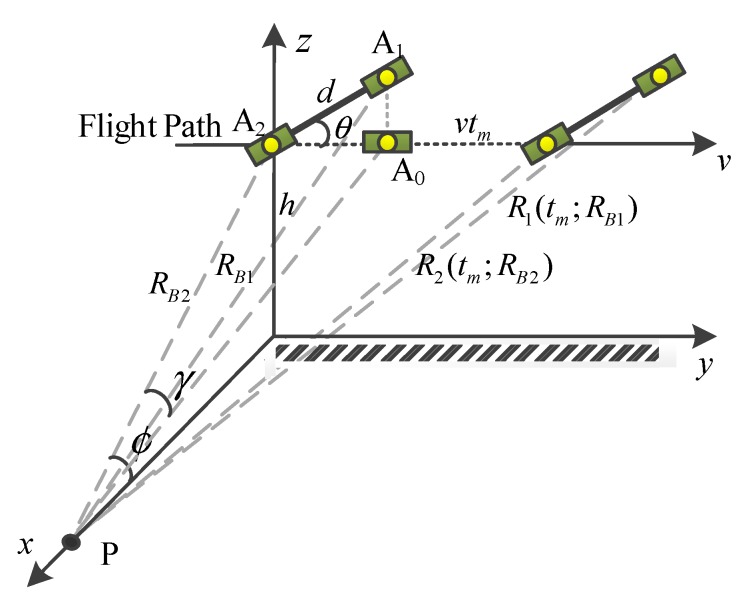
The imaging geometry for an airborne dual-channel CSAR/GMTI system.

**Figure 2 sensors-20-00158-f002:**
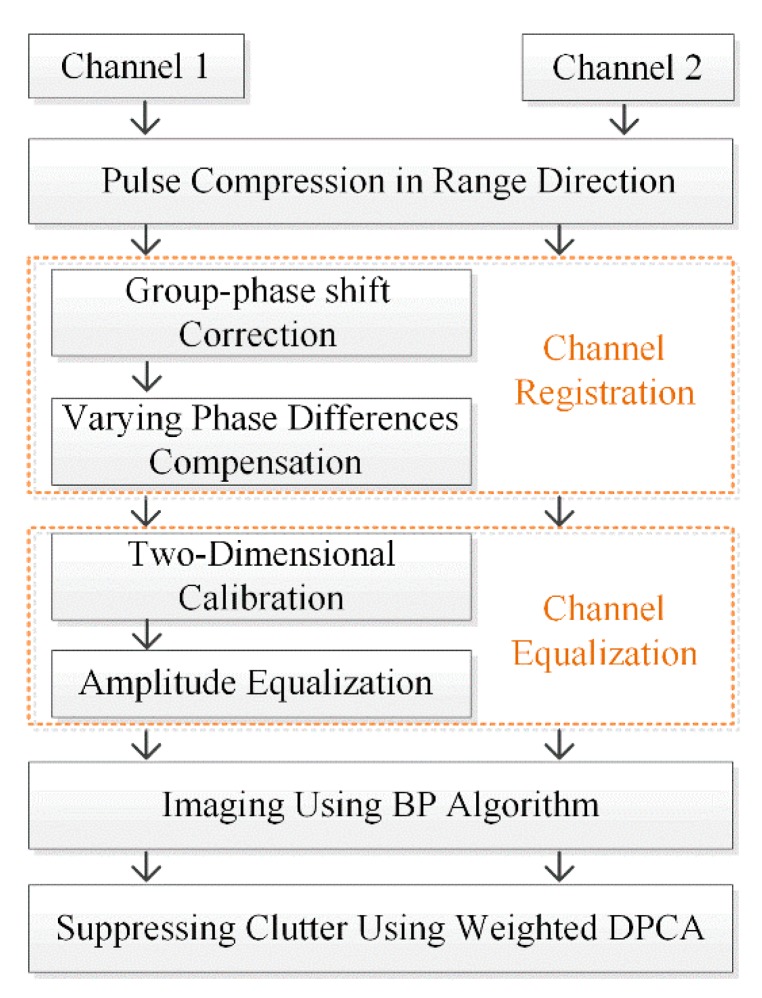
The flowchart of a dual-channel circular synthetic aperture radar (CSAR) ground moving target indication (GMTI) method. BP: back-projection, DPCA: displaced phase center antenna.

**Figure 3 sensors-20-00158-f003:**
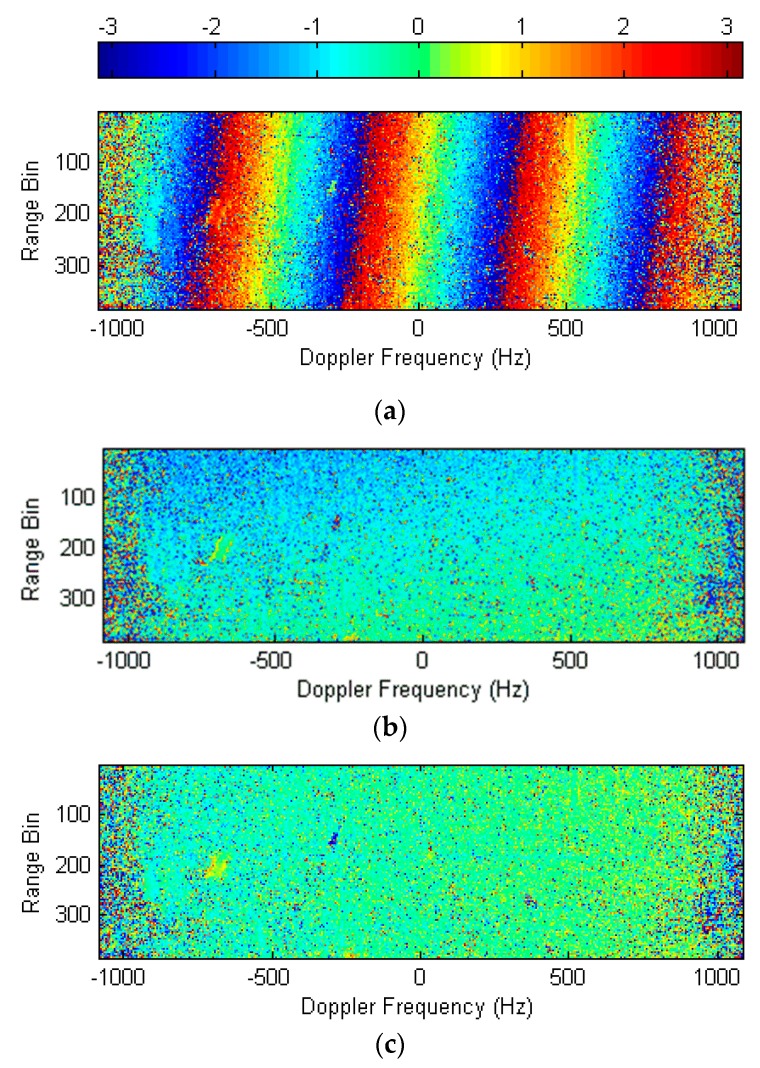
Interferometric phase diagrams in the range Doppler domain. (**a**) The interferometric phase diagram between the original dual-channel signals. (**b**) The interferometric phase diagram after the linear phase compensation. (**c**) The interferometric phase diagram after the group-phase shift correction.

**Figure 4 sensors-20-00158-f004:**
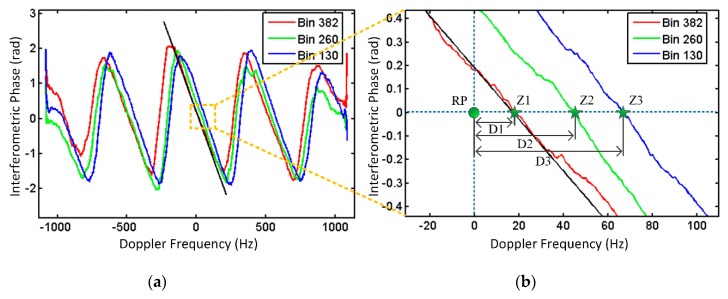
Curves for the three range bins. (**a**) The phase curves of the 130th, 260th, and 382nd range bins. (**b**) The partially enlarged curves of (**a**).

**Figure 5 sensors-20-00158-f005:**
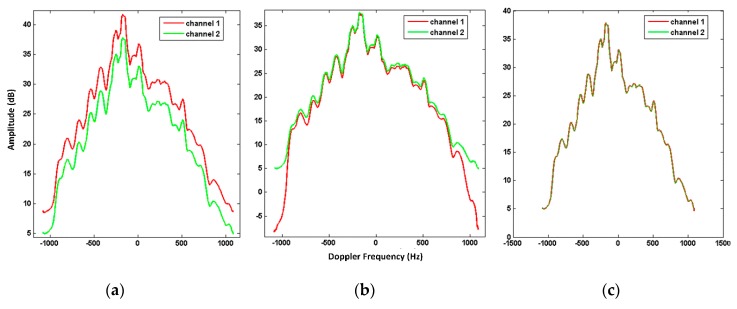
The amplitude curves of the dual-channel signals: (**a**) after channel registration, (**b**) after 2D calibration, and (**c**) after 2D calibration and amplitude equalization.

**Figure 6 sensors-20-00158-f006:**
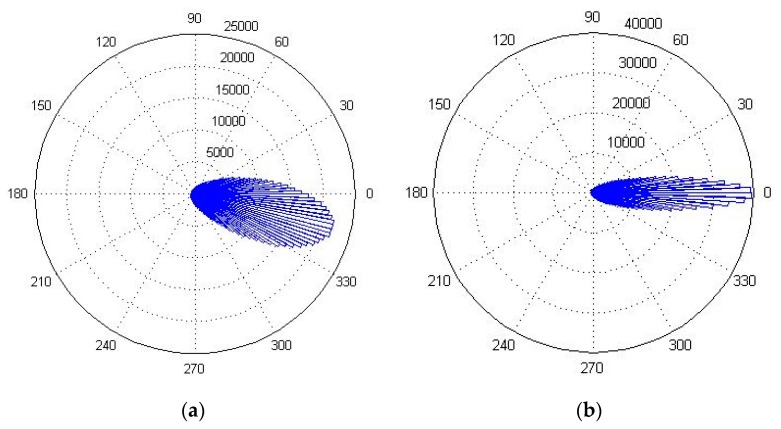
The statistical angle histograms of the interferometric phase: (**a**) before channel equalization and (**b**) after channel equalization.

**Figure 7 sensors-20-00158-f007:**
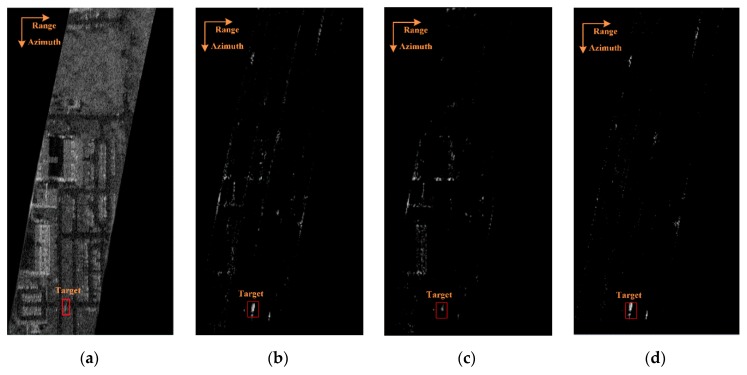
The clutter suppression effect. (**a**) Original SAR image. Clutter suppression results with (**b**) $\xi_{1}$, (**c**) $\xi_{2}$, and (**d**) $W_{DPCA}$.

**Figure 8 sensors-20-00158-f008:**
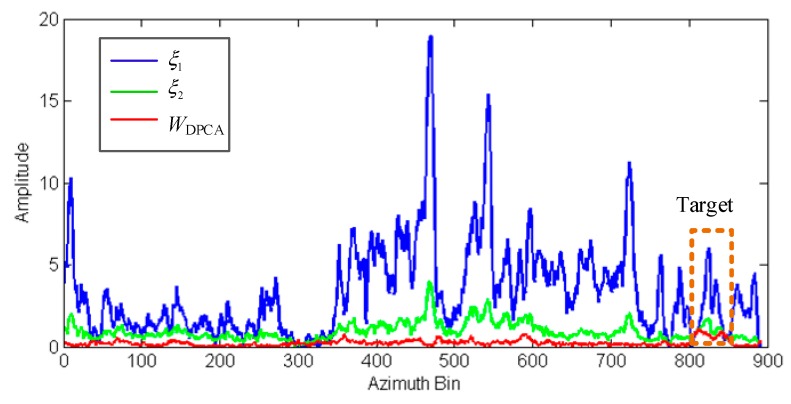
The amplitude curves of SAR images manipulated with $\xi_{1}$, $\xi_{2}$, and $W_{DPCA}$.

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
