# Peer review of "Moving Target Indication for Dual-Channel Circular SAR/GMTI Systems"

_sensors, 2019, doi:10.3390/s20010158_

Round 1
Reviewer 1 Report
The paper proposes a signal processing procedure for MTI for a dual-channel CSAR.GMTI system. The procedure mainly includes channel registration, channel equalization, weighted DPCA clutter suppression. The effectiveness of the method is demonstrated through an experiment with real data. The paper can be considered for publication. However, I just have some comments as follows:
1. The introduction section does not have enough reference to more recent research work on the same topic. Please add. Moreover, references [6] [7] [8] are not referred in the paper so please consider dropping them. [10] and [11] should be in the Introduction too.
2. The proposed method looks very similar to the authors' previous work presented at the 14th International Conference on Wireless Communications, Networking and Mobile Computing (WiCOM 2018) with the title "A Clutter Suppression Method for Dual-channel Circular SAR-GMTI".
Please highlight the new details and new results in the current manuscript.
3. It would be nice if the proposed method can be validated with different datasets rather than just one.
4. A few minor grammar errors that I can spot:
In the Introduction section, "which employing" should read "which employs" On line 218 (section 3.3), "In one reference [10]" should be "In reference [10]" On line 219 (section 3.3), "In the other one [11]" can be changed to "In reference [11]" In Section 4 (Conclusions), "we present a complete data procedure" should read "we present a complete data processing procedure"Please revise the manuscript thoroughly.
Reviewer 2 Report
This paper proposes a method for channel imbalance compensation in multiple channel SAR system. The proposal sounds somehow reasonable however; the authors must re-write the manuscript in the proper format prior to be reviewed.
1. The authors wrote the proposed method and its experimental results in the same section step by step and thus, it is difficult to understand the theoretical expectation and the practical results. In addition, there is no numerical comparison between the conventional methods.
2. From Section 3.1, it is assumed that the authors expect that there is only one specific target in the specific range window. On the other hand, there should be multiple targets with multiple “k” values in general case. How did the authors deal with it?
3. The authors have to cite the proper reference for the dataset.
4. Reference list is incomplete. 3 out of 11 references are Ph. D. theses and 2 are technical report. That is, half of the references are non-reviewed paper.
Round 2
Reviewer 1 Report
The revised version looks good to me. Thanks for explaining the difference and adding new results. I just have a few minor suggestions.
Please refer to your WiCOM 2018 conference paper as your previous work, as well as add the explanation of difference to the Introduction section. Since you added a new dataset (Ku band), it should be mentioned at the beginning of Section 4. For example, what is the source of the dataset? or something similar to the first dataset. In Figure 9a, please also include zoomed in pictures of target 1 and 2 as in Figure 9b. I think a more detailed analysis of the new dataset (similar to the 1st dataset) would be useful.Author Response
Please see the attachment.

Reviewer 2 Report
The revised paper seems mostly fine. Some minor problems remain.
1. Channel1 cannot be seen in Fig. 6(c). In order to increase the visibility, please change the width of the lines or use dotted line to show both lines instead.
2. Use specific term instead of "anther dataset" for the title of Section 4.4. What is the topic of this section?
In addition, please describe further specifically what are the difference and purpose of this dataset. What is the expected difference? How was the result? There is no specific description for the evaluation results.
